# PSME4 Degrades Acetylated YAP1 in the Nucleus of Mesenchymal Stem Cells

**DOI:** 10.3390/pharmaceutics14081659

**Published:** 2022-08-09

**Authors:** Yong Sook Kim, Mira Kim, Dong Im Cho, Soo Yeon Lim, Ju Hee Jun, Mi Ra Kim, Bo Gyeong Kang, Gwang Hyeon Eom, Gaeun Kang, Somy Yoon, Youngkeun Ahn

**Affiliations:** 1Biomedical Research Institute, Chonnam National University Hospital, Gwangju 61469, Korea; 2Department of Pharmacology, Chonnam National University Medical School, Hwasun 58128, Korea; 3Division of Clinical Pharmacology, Chonnam National University Hospital, Gwangju 61469, Korea; 4College of Pharmacy, Chonnam National University, Gwangju 61186, Korea; 5Department of Cardiology, Chonnam National University Hospital, Gwangju 61469, Korea

**Keywords:** mesenchymal stem cell, acetylation, YAP1, PSME4

## Abstract

Intensive research has focused on minimizing the infarct area and stimulating endogenous regeneration after myocardial infarction. Our group previously elucidated that apicidin, a histone deacetylase (HDAC) inhibitor, robustly accelerates the cardiac commitment of naïve mesenchymal stem cells (MSCs) through acute loss of YAP1. Here, we propose the novel regulation of YAP1 in MSCs. We found that acute loss of YAP1 after apicidin treatment resulted in the mixed effects of transcriptional arrest and proteasomal degradation. Subcellular fractionation revealed that YAP1 was primarily localized in the cytoplasm. YAP1 was acutely relocalized into the nucleus and underwent proteasomal degradation. Interestingly, phosphor-S127 YAP1 was shuttled into the nucleus, suggesting that a mechanism other than phosphorylation governed the subcellular localization of YAP1. Apicidin successfully induced acetylation and subsequent dissociation of YAP1 from 14-3-3, an essential molecule for cytoplasmic restriction. HDAC6 regulated both acetylation and subcellular localization of YAP1. An acetylation-dead mutant of YAP1 retarded nuclear redistribution upon apicidin treatment. We failed to acquire convincing evidence for polyubiquitination-dependent degradation of YAP1, suggesting that a polyubiquitination-independent regulator determined YAP1 fate. Nuclear PSME4, a subunit of the 26 S proteasome, recognized and degraded acetyl YAP1 in the nucleus. MSCs from PSME4-null mice were injected into infarcted heart, and aberrant sudden death was observed. Injection of immortalized human MSCs after knocking down PSME4 failed to improve either cardiac function or the fibrotic scar area. Our data suggest that acetylation-dependent proteasome subunit PSME4 clears acetyl-YAP1 in response to apicidin treatment in the nucleus of MSCs.

## 1. Introduction

Cardiovascular disease (CVD) such as myocardial infarction (MI) is a major cause of global death worldwide [1]. Currently, there is an unmet need for novel therapeutics to relieve cardiac fibrosis, inflammation, and ventricular remodeling. Stem cell-based therapy has remarkable promise for repair of cardiac injury. In particular, mesenchymal stem cells (MSCs) have produced encouraging results for a variety of pathological conditions, including MI, in both preclinical studies and clinical trials [2,3,4,5,6].

MSC-based regeneration therapies are currently used to treat both acute MI and chronic ischemic cardiomyopathy [7]. The suggested mechanisms via which MSCs induce cardiac regeneration include angiogenesis, anti-inflammation, and antifibrosis, resulting in cardiac repair after MI [8,9,10]. Although priming or genetic modification has been developed to enhance the therapeutic effects of MSCs, the rate of MSC differentiation into cardiomyocytes or cardiomyocyte-like cells is hardly improved compared with embryonic stem cells and induced pluripotent stem cells. Numerous strategies have been developed to promote the cardiac differentiation of MSCs. For example, miRNA1-2 overexpression, 5-azacytidine, basic fibroblast growth factor, and Jagged 1 are known to induce the phenotypic differentiation of MSCs into cardiomyocyte-like cells [11,12,13].

In 2010, Terzic et al. showed that treatment of MSCs with a cocktail of TGF-β, BMP-4, Activin-A, IGF-1, IL-6, FGF2, thrombin, and retinoic acid could successfully upregulate cardiac transcription factors to facilitate cardiopoiesis [14]. Cardiopoietic MSCs have subsequently been tested in clinical trials, such as the prospective multicenter phase 2 C-CURE trial (cardiopoietic stem cell therapy in heart failure), in which lineage-specified MSCs were injected via the endomyocardial route and showed beneficial effects in patients with ischemic heart failure [15].

The follow-up phase 3 CHART-1 trial (congestive heart failure cardiopoietic regenerative therapy) was a double-blind study designed to assess cardiac outcomes in ischemic heart failure patients treated with endomyocardial injection of lineage-specified MSCs. Two years of clinical data demonstrated safety, but the treatment did achieve the primary endpoint compared with sham control. Several questions—including cell dosage, delivery route, and cell state—remain unsolved to meet therapeutic efficacy. Therapeutic benefit was observed in the heart failure with cardiac dilation group [16].

We previously showed that apicidin treatment induces early cardiac gene expression by reducing YAP1 (Yes-associated protein 1) in MSCs [17]. YAP1 is a transcriptional co-activator protein that shuttles between the cytosol in its phosphorylated inactive state and the cell nucleus in its unphosphorylated active state [18]. The target genes of YAP1 are regulated by binding interactions with TEAD (TEA/ATTS domain) transcription factors. YAP1 localization, regulated by mechanical environment, matrix stiffness, and biochemical cues, has been reported to be required for the lineage specification of MSCs. We found significant downregulation of YAP1 by apicidin in MSCs. In this study, we identified PSME4 as a novel modifier of nuclear YAP1 protein that functions to degrade acetyl-YAP1 in the MSCs [19].

## 2. Materials and Methods

### 2.1. Antibodies and Reagents

Antibodies used were as follows: anti-YAP1, anti-phosphor-YAP1 (S127), and anti-acetyl-lysine were from Cell Signaling Technology (Danvers, MA, USA). Anti-PSME4, anti-histone H3, and anti-HDAC2 were from Abcam (Cambridge, UK). Anti-14-3-3 epsilon was from Thermo Fisher Scientific (Waltham, MA, USA). Anti-HSP90, anti-HDAC6, anti-acetylated-tubulin, and anti-GAPDH were from Santa Cruz Biotechnology (Dalla, CA, USA). Anti-actin, anti-tubulin, anti-HA, and anti-flag were from Sigma (St. Louis, MO, USA). Alexa Fluor 568-conjugated goat anti-mouse IgG was from Molecular Probes (Invitrogen, Waltham, MA, USA).

Reagents used were as follows: apicidin, cycloheximide, chloroquine, actinomycin D, leptomycin B, nicotinamide, tubastatin A, and doxycycline were from Sigma (St. Louis, MO, USA). MG132 was purchased from Cayman (Ann Arbor, MI, USA). Puromycin and blasticidin were from InvivoGen (San Diego, CA, USA).

siRNAs targeting HDAC6, FBXW7, SOCS6, and PSME4 were purchased from Dharmacon (Lafayette, CO, USA). Scramble siRNA was purchased from Bioneer (Daejeon, Korea).

### 2.2. Immortalized Human Mesenchymal Stem Cells

Immortalized human bone marrow (BM)-derived MSCs (hTERT-MSCs) utilized for the study were kindly provided by Professor Yeon-Soo Kim (Inje University, Inje, Korea). Immortalization was carried out by infecting the cells with telomerase and selecting with puromycin [20]. hTERT-MSCs were cultured in Dulbecco’s modified Eagle’s medium with 10% fetal bovine serum and 1% penicillin/streptomycin (Hyclosne, UT, USA). Cells were grown at 37 °C in a humidified atmosphere of 95% air and 5% CO_2_. hTERT-MSCs express CD90 but are negative for CD45, CD34, CD14, and CD11b. Differentiation properties into adipogenic, osteogenic, or chondrogenic lineages were previously confirmed [17].

### 2.3. Small Hairpin RNA

To knock down endogenous gene expression, lentivirus-driven small hairpin (sh) RNA was utilized. Desired nucleotides were cloned into the reverse tetracycline transactivator (rtTA) system-controlled shuttle vector (LT3GEPIR), and pMD2.G and psPAX2 were used for generation of lentivirus. Lentiviral infection and shRNA activation were monitored by brief checking for green fluorescent protein (GFP) with 2 μg/mL of doxycycline. Knockdown efficiency was checked by Western blot or quantitative real-time PCR 2 days after doxycycline treatment. Nucleotide sequences utilized were as follows:

Nontargeting control: 

5′-TGCTGTTGACAGTGAGCGCAGGAATTATAATGCTTATCTATAGTGAAGCCACAGATGTATAGATAAGCATTATAATTCCTATGCCTACTGCCTCGGA-3′,

Human PSME4: 

5′-TGCTGTTGACAGTGAGCGCTCAGAAGATGATACTAAGTCATAGTGAAGCCACAGATGTATGACTTAGTATCATCTTCTGAATGCCTACTGCCTCGGA-3′.

### 2.4. Subcellular Fractionation

Subcellular fractions of cytoplasm and nucleus were prepared by manual manipulation. Cells were dissolved and incubated for 5 min on ice with hypotonic buffer (10 mM HEPES pH 7.9, 10 mM KCL, 1 mM DTT, and protease inhibitor mixture (Gendepot, TX, USA, P3100)). After incubation, Igepal CA-630 was added to a final concentration of 0.6%, and samples were thoroughly mixed for 10 s. After centrifugation at 16,000 RCF for 30 s, the supernatant was saved for the cytoplasm fraction. The pellet was washed twice with cold PBS, and hypertonic buffer (20 mM HEPES pH 7.9, 400 mM NaCl, 1 mM DTT, protease inhibitor mixture) was added. The nuclear fraction was extracted by incubation for 15 min on ice and acquired by centrifugation at 16,000 RCF for 10 min. Supernatant was captured for the nuclear fraction. HSP90 or GAPDH was utilized for visualization of the cytoplasmic fraction, and HDAC2 or Histone H3 was utilized for visualization of the nuclear fraction.

### 2.5. Phos-Tag Gel

To exaggerate and clarify the phosphorylation of YAP1, we used SDS-PAGE with Phos-tag phosphoprotein gel stain (WAKO Chemical, Osaka, Japan). The stained gels were made by following the manufacturer’s instructions. Briefly, both 50 μM Phos-tag and 100 μM MnCl_2_ were additionally mixed into a standard 6% SDS-PAGE gel formula. Before transfer to the membrane, the phosphoprotein stain gel was incubated with 10 mM EDTA containing transfer buffer to chelate MnCl_2_ for 10 min, which was repeated three times. The gel was then washed with EDTA-free transfer buffer for 10 min, and proteins were transferred to membranes for regular Western blot.

### 2.6. Immunoprecipitation

For immunoprecipitation, 1 mg of cell lysate was prepared with 1% NP buffer (1% Igepal CA-630, 50 mM Tris-HCl pH 8.0, 150 mM NaCl, 1 mM EDTA, and protease inhibitor mixture). One microgram of primary antibody was added and allowed to probe for 2 h with continuous rotation at 4 °C. Ten microliters of protein A/G magnetic beads (LSKMAGAG02, Millipore, Burlington, MA, USA) were added to collect antibodies. After 2 h of continuous rotation at 4 °C, beads were precipitated with a magnet and washed twice with 1% NP buffer. The precipitants were denatured and reduced by boiling for 5 min after adding NuPAGE SDS sample buffer (Invitrogen, Waltham, MA, USA, NP0007) with beta-mercaptoethanol (Sigma, St. Louis, MO, USA, #63689). Proteins were separated by SDS-PAGE, and 30 μg of protein was utilized for input control.

### 2.7. Immunocytochemistry and Cell Imaging

Cells were prepared on cover slips and maintained in six-well plates. Specifically conditioned cells were fixed with 3.7% (*v*/*v*) paraformaldehyde for 10 min and then washed with PBS. Permeabilization was carried out with 0.5% (*v*/*v*) Triton X-100 containing PBS for 10 min with continuous agitation. After brief washing with PBS three times for 5 min, blocking proceeded with PBS containing 1% (*w*/*v*) BSA (1% BSA/PBS) for 1 h at room temperature. After blocking with BSA, cells were incubated overnight with primary antibodies (Flag, 1:1000) in blocking buffer (1% BSA/PBS). Primary antibodies were washed with PBS containing 0.1% (*v*/*v*) Tween-20 (0.1% PBS/Tween-20) three times. Flag antibodies were further probed by Alexa-conjugated secondary antibodies (568 for mouse). Cover slips were flipped and mounted with one drop of antifade solution containing 6-diamidino-2-phenylindole (DAPI, Molecular Probes, Eugene, OR, USA). Cells were analyzed using the NIS-Elements AT program (Nikon Inc., Tokyo, Japan). Enhanced GFP (EGFP)-fused YAP1 was briefly fixed and directly imaged without any staining.

### 2.8. Quantitative Real-Time Polymerase Chain Reaction

Total mRNA was extracted with TRIzol (Invitrogen, Waltham, MA, USA, #15596026). cDNA was synthesized by use of random hexamer (M-MLV reverse transcriptase, Invitrogen, Waltham, MA, USA, #28025013). Quantitative real-time PCR was carried out by using QuantiTect SYBR Green kits (Qiagen, Hilden, Germany, #204143) with a Rotor-Gene Q (Qiagen, Hilden, Germany). PCR analysis was performed in triplicate, and the average was regarded as a single result. The relative contents of mRNA transcripts were normalized to those of GAPDH. Primer sets for human YAP1 and PSME4 were purchased (Bioneer, Daejeon, Korea, P245469V for YAP1 and P208885V for PSME4). Specific oligomer sets designed were as follows:

Human GAPDH, sense: 5′-GTCTCCTCTGACTTCAACAGCG-3′, antisense: 5′-ACCACCCTGTTGCTGTAGCCAA-3′

### 2.9. TEAD Reporter Activity

Luciferase activity was measured to check TEAD transcriptional activity by using commercial kits (GloMax^®^, Promega, Madison, WI, USA) according to the manufacturer’s instructions. Reporter plasmid with eight copies of wildtype TEAD-binding sequences was kindly provided by Jong In Yook (Yonsei University, Seoul, Korea). Two days after transfection, cells were lysed in reporter lysis buffer. TEAD activity was measured with a luminometer (Glomax^®^, Promega, Madison, WI, USA). *Renilla* luciferase was utilized for normalization.

### 2.10. Animal Model

The animal usage for disease models was approved by the Chonnam National University Medical School Research Institutional Animal Care and Use Committee (CNU IACUC-H-2020-12). For the MI model, 8 week old male Balb/C nude mice underwent permanent ligation of the coronary artery. Mice were anesthetized with an intramuscular injection of ketamine (50 mg/kg) and xylazine (5 mg/kg) and maintained with an artificial ventilator. The intercostal space was widened, and the proximal part of the left anterior descending coronary artery was ligated with 7-0 silk thread. For cell transplantation, 5 × 10^5^ rtTA-shControl hTERT-MSCs, rtTA-shPSME4 hTERT-MSCs, WT BM-MSCs, or PSME4 KO BM-MSCs cells were prepared in 100 μL of PBS. Knockdown of PSME4 in hTERT-MSCs was transduced by doxycycline (2 μg/mL) 1 day before transplantation. Direct injection of vehicle or conditioned cells was performed into random fields throughout the left-ventricular (LV) myocardium after ligation of the coronary artery. After 2 weeks, the ejection fraction and fractional shortening were analyzed by echocardiography, and the animals were euthanized by use of carbon dioxide chamber for further analysis. Survival rates were calculated using the Kaplan–Meier method and were visualized using Prism 9.2 (GraphPad, San Diego, CA, USA). Survival curves were compared using the Mantel–Cox test. Significance was determined at values of *p* < 0.05.

### 2.11. Histology

Cardiac fibrosis was assessed with Masson’s trichrome staining. The heart tissues were harvested, fixed in formalin, and embedded in paraffin blocks at 14 days after MI. The sections were cut at 6 μm, and the slides were stained with Masson’s trichrome staining kit according to the manufacturer’s instructions (Abcam, Cambridge, UK, ab150686). Fibrotic areas were measured by visualizing blue staining by using the NIS-Elements Advanced Research program (Nikon, Tokyo, Japan). The percentage of fibrosis was calculated as the blue-stained area divided by the LV area, and the cross-sectional wall thickness of the scar adjacent to border zones was measured.

### 2.12. Genetically Engineered Mice

PSME4 knockout mice were purchased from RIKEN Bioresource Research Center (RBRC09401, Wako, Japan) after permission was received from the original depositor, Professor Tomoki Chiba (University of Tsukuba, Tsukuba, Japan) [21]. The oligomer set for genotyping was as follows: 

Reverse antisense: 5′-GAGACCTTCTGCACTTCCAAGGATCTCAT-3′,

Sense: 5′-CCTCCCAAGTGTCTAAAGCCGCTTATACTG-3′,

Sense(neo): 5′-TCGTGCTTTACGGTATCGCCGCTCCCGATT-3′.

Neomycin cassette produced 1 kb bands while the wildtype generated 600 base pairs of products.

### 2.13. Echocardiography

Cardiac function was measured by ultrasonography (Vivid S5, General Electric Company, Boston, MA, USA). Mice were anesthetized with intramuscular injection of ketamine (50 mg/kg)/xylazine (5 mg/kg), and ultrasound gel was applied to the chest after it was shaved. At the papillary muscle level, two-dimensional M-mode was acquired from the parasternal long-axis view or parasternal short-axis view. Ejection fraction was determined using the Teichholz formula, EF (%) = (Vd − Vs)/Vd, where Vd indicates LV volume at end diastole and Vs is at end systole, and Vd = [7/(2.4 + LVIDd)] × LVIDd^3^, Vs = [7/(2.4 + LVIDs)] × LVIDs^3^, where LVIDd is LV interventricular dimension at end diastole and LVIDs is that at end systole. Fractional shortening was measured by the formula FS = (LVIDd − LVIDs)/LVIDd.

### 2.14. Statistics

Statistical significance was analyzed with PASW Statistics 27 (SPSS, IBM Corp., Armonk, NY, USA). One-way analysis of variance (ANOVA) with post hoc multiple comparison was used. When there were two or more main effects, two-way ANOVA was applied. If an interaction of the main effects was confirmed as being significant, further stratification was carried out to perform pairwise comparison. For post hoc tests, Tukey’s HSD (honestly significant difference) was applied for multiple comparisons in equal variance.

## 3. Results

### 3.1. Apicidin Triggers Nuclear Redistribution of YAP1 and Promotes Proteasomal Degradation in the Nucleus

We reported previously that overnight treatment of primary cultures of MSCs or immortalized human MSCs (hTERT-MSCs) with apicidin induces acute ablation of YAP1 and subsequent early cardiac gene expression [17]. Here, we checked how rapidly YAP1 is lost after apicidin exposure. We observed a decrease in YAP1 protein as early as after 6 h of apicidin treatment (Figure 1a). Overnight treatment strongly arrested the transcriptional activity of YAP1, which is known to be through transactivation of p21, a well-known target of histone deacetylase (HDAC) inhibitor. As reported, acute transcriptional arrest upon apicidin treatment was confirmed in hTERT-MSCs (Appendix A). Interestingly, apicidin failed to regulate TAZ. TAZ still remained in the cytoplasm unchanged (Appendix A).

We next questioned whether apicidin solely regulated the transcription of YAP1 or whether another mechanism was also involved. De novo protein synthesis was halted by overnight treatment with either cycloheximide (CHX, translational inhibitor) or actinomycin D (ActD, transcription inhibitor), and YAP1 protein was measured. The natural clearance of YAP1 in hTERT-MSCs after CHX or ActD treatment was relatively slower than after apicidin treatment (Figure 1b). We further tested this effect with higher concentrations of CHX and ActD. Clearance of YAP1 after overnight treatment was faster in the apicidin-treated group than after treatment with CHX or ActD (Appendix A), suggesting that apicidin might induce another form of regulation simultaneously, such as active protein degradation. 

We next examined the two main pathways for protein degradation: the lysosome (or autophagy) pathway and the proteasome pathway. Chloroquine, which functions in the lysosome pathway, failed to alleviate YAP1 degradation (Figure 1c). Next, we tested the proteasomal pathway with MG132. hTERT-MSCs were incubated with apicidin overnight, and MG132 was added to the apicidin-containing media 8 h before harvest. MG132 successfully reduced YAP1 degradation. More interestingly, MG132 alone greatly increased the YAP1 level, which indicates that the proteasome pathway also regulates spontaneous turnover of YAP1 even in the resting state of hTERT-MSCs (Figure 1d). 

Next, we questioned in which subcellular fraction YAP1 was degraded. We briefly fractionated the subcellular component into cytoplasm and nucleus after MG132 treatment, and the fractions were tested with Western blot. Restored YAP1 accumulated predominantly in the nucleus, whereas MG132 alone tended to halt degradation of YAP1 in the cytoplasm (Figure 1e). Because treatment with MG132 increased YAP1 in both the cytoplasm and the nucleus, we further tested subcellular redistribution during proteasomal degradation. Nuclear export was blocked by treatment with leptomycin B (LMB), a CRM1 inhibitor which governs nuclear export. Treatment with MG132 alone resulted in the accumulation of YAP1 both in the cytoplasm and in the nucleus (first vs. second and seventh vs. eighth of Figure 1f), but further increases of YAP1 in the nucleus were not observed (second vs. sixth and eighth vs. 12th of Figure 1f). Similarly, acute degradation of YAP1 was not affected by treatment with LMB; cytoplasmic YAP1 was not decreased by LMB (third vs. fourth of Figure 1f) and nuclear YAP1 was not increased even in the presence of LMB (ninth vs. 10th of Figure 1f), suggesting that apicidin promoted unidirectional redistribution from the cytoplasm to the nucleus. In apicidin-treated MSCs, YAP1 underwent proteasome-dependent degradation in the nucleus followed by nuclear redistribution.

### 3.2. Posttranslational Modifications Other Than S127 Phosphorylation Regulate YAP1 Shuttling

Among the numerous posttranslational modifications of YAP1, the mechanism of S127 phosphorylation is well established; Lats1/2 phosphorylates S127 of YAP1, which leads to 14-3-3 recognition and, thus, cytoplasmic restriction [22]. Hence, we first checked whether apicidin modulated YAP1 S127 phosphorylation indirectly during nuclear localization. The phosphoprotein stain gel successfully separated phosphorylated and unphosphorylated proteins by exaggerated retardation of phosphorylated forms during SDS-PAGE. As in previous reports [23], total YAP1 antibody visualized widely separated cytoplasmic YAP1. However, we found that nuclear YAP1 was also shifted (Figure 2a, bottom panel). We then used a phosphor-S127-specific antibody to confirm the identity of the shifted YAP1 retardation. As expected, phosphorylation of YAP1 S127 was mostly detected in the cytoplasmic fraction. However, phosphor-S127 YAP1 antibody confirmed that nuclear YAP1 was also phosphorylated at S127 (Figure 2a, top panel). We further figured out the role of phosphorylation of S127 during YAP1 shutting upon apicidin treatment. As shown Figure 1, apicidin promoted nuclear localization of YAP1 (Appendix A), and redistributed YAP1 upon apicidin stimulation was positive against phosphor-S127 YAP1 antibodies (first vs. second and fifth vs. sixth, Figure 2b), which indicated that a mechanism other than phosphorylation governs subcellular localization of YAP1.

### 3.3. Acetylation Participates in Subcellular Localization of YAP1

Apicidin is a potent HDAC inhibitor (HDACi) that regulates acetylation of histone or nonhistone proteins [24]. Hence, we postulated the involvement of YAP1 acetylation in response to apicidin treatment. We checked the acetylation of YAP1 after 2 h of treatment with apicidin to avoid possible degradation of YAP1. Brief treatment with apicidin successfully induced YAP1 acetylation, which resulted in dissociation of 14-3-3 from YAP1 (Figure 3a). In addition to YAP1, 14-3-3 acetylation was further examined but we failed to acquire convincing evidence of 14-3-3 acetylation (Appendix A). To elucidate the acetylation locus, we performed Western blot after subcellular fractionation. Two hours of treatment with apicidin induced YAP1 acetylation in the cytoplasmic fraction. Interestingly, acetylated YAP1 was predominantly detected in the nucleus (Figure 3b). Given that Figure 1 showed the unidirectional redistribution of YAP1 from the cytoplasm to the nucleus, acetylation of YAP1 in the cytoplasm might precede shuttling into the nucleus. Hence, we hypothesized that cytoplasmic HDAC might regulate YAP1 acetylation.

To further determine the subtype of HDAC(s) involved in YAP1 deacetylation, we tested YAP1 degradation by treatment with various HDACi including apicidin. Among those tested, tubastatin A (TST), a selective HDAC6 inhibitor [25], successfully degraded YAP1 as strongly as did apicidin, whereas nicotinamide, a potent sirtuin family inhibitor, failed to do so (Figure 3c). Total acetylation of YAP1 was also induced after 2 h of incubation with TST (Appendix A). We further confirmed that overnight treatment with TST strongly arrested YAP1 transcription (Appendix A).

As noted, TST is a specific HDAC6i. To specify the role of HDAC6 in YAP1 regulation, endogenous HDAC6 was specifically targeted by use of siRNA transfection. As with overnight treatment with apicidin or TST, total YAP1 was significantly reduced by 3 days after siRNA transfection (Figure 3d). Interestingly, nuclear localization of YAP1 was increased within 1 day after siRNA transfection (Figure 3e), which was a phenocopy of the effect of a few hours of treatment with apicidin [17]. We further checked whether another subtype of HDAC might involve YAP1 acetylation and, hence, degradation after shuttling into the nucleus. We removed endogenous HDAC6 using siRNA and then treated the cells with apicidin or TST. Apicidin or TST itself successfully reduced YAP1 in the nontargeting siRNA condition, but no further decrease in YAP1 was observed when cells were transfected with siRNA against HDAC6 (Figure 3f). We measured TEAD reporter activity in response to nuclear YAP1 using an artificial promoter and observed that the Hippo pathway was activated by either HDACi or HDAC6 siRNA (Figure 3g). Hence, we concluded that HDAC6 was necessary and enough to regulate YAP1 acetylation.

### 3.4. YAP1 Lysine 494/497 Are Acetylation Targets

To clarify the role of acetyl-YAP1, we mutated candidate residues, lysine 494 and lysine 497, according to a previous reference [26]. As reported, total acetylation of YAP1 was not observed when lysines 494/497 were mutated (Figure 4a). We showed that YAP1 undergoes acetylation in response to apicidin treatment (Figure 3), nuclear translocation, and proteasomal degradation in the nucleus (Figure 1). Subcellular localization of the acetyl-dead mutant of YAP1 was visualized by immunofluorescence with Flag antibody (Appendix A) or shown directly by enhanced green fluorescent protein (EGFP) fused at the C terminus of YAP1 (Figure 4b). Incubation with apicidin for 5 h was enough to enrich wildtype YAP1 in the nucleus. By contrast, the acetyl-dead mutant of YAP1 was restricted to the cytoplasm (Figure 4b and Appendix A). We directly counted cell numbers showing an EGFP signal in the nucleus and divided by the total number of cells in random fields. The nuclear EGFP-positive population, which represents nuclear shuttling from the cytoplasm, was significantly lower when two lysines were mutated (Figure 4c). In agreement with Figure 2, S127 phosphorylation of YAP1-EGFP was not altered by acetylation status (Appendix A). We further questioned whether the acetyl-dead mutant was resistant to apicidin stress because it failed to promote nuclear localization (Figure 4b,c). Apicidin, indeed, failed to reduce the acetylation-dead mutant of YAP1, whereas wildtype YAP1 was significantly decreased (Figure 4d). 

### 3.5. PSME4 Degrades Acetyl-YAP1 in the Nucleus

So far, we delineated that apicidin induced YAP1 acetylation by inhibiting cytoplasmic HDAC6 (Figure 3) and then promoted nuclear localization (Figure 3 and Figure 4), which resulted in degradation of YAP1 in the nucleus (Figure 1 and Figure 4). Because MG132 blocked degradation of nuclear YAP1 (Figure 1), we tested the role of the well-known E3 ligases of YAP1 in the hTERT-MSCs. Although siRNA successfully targeted each E3 ligase (Appendix A), YAP1 degradation by overnight treatment with apicidin was not affected (Figure 5a, second, fourth, and sixth lanes, Appendix A). As shown above, the proteasomal inhibitor MG132 obviously preserved YAP1, especially in the nucleus. However, no significant increase in polyubiquitination was observed (Figure 5b), even though the overall polyubiquitination pathway was dramatically accelerated in the presence of apicidin (Appendix A). Hence, we searched for polyubiquitination-independent proteasome regulators. We tested the nuclear proteasome member, proteasome activator subunit 4 (PSME4, a subunit of 26S proteasome), for its bromodomain-like domain that recognizes and binds to acetyl protein [21]. Apicidin-dependent degradation of YAP1 was attenuated when cells were transfected with siRNA against PSME4 (Figure 5a 2nd vs. 8th). 

Next, we generated rtTA-shPSME4 that bound to endogenous PSME4 and confirmed efficiency in infected cells (Appendix A). Targeted deletion of PSME4 attenuated YAP1 degradation upon apicidin treatment (Figure 5c, fourth vs. eighth lane). As expected, preserved YAP1 in rtTA-shPSME4-infected cells accumulated mostly in the nucleus (Figure 5d). 

### 3.6. Programmed Regulation of Nuclear YAP1 by PMSE4

Next, we introduced PSME4-null mice to clarify its role in YAP1 regulation. As reported, PSME4 homozygous males were infertile as reported [21], and overall heart size was relatively smaller in null mice. Otherwise, we could not identify apparent alterations in homozygous mice (Appendix A). We isolated bone marrow (BM) MSCs from PSME4 knockout mice (PSME4-KO MSCs) or wildtype littermates and confirmed molecular changes. As expected, YAP1 protein was more abundant in PSME4 KO MSCs than in wildtype littermates (Figure 6a). Interestingly, transcription activity of YAP1 was already downregulated in PSME4-KO MSCs, and further suppression by apicidin treatment was not observed (Figure 6b). To delineate the role of PSME4 in MI, we directly injected MSCs obtained either from PSME4-KO or wildtype littermates to infarcted heart in a MI mouse model and monitored the survival rate for 2 weeks. As repeatedly reported, delivery of MSCs significantly improved total survival in the 2 week cohort study. However, we experienced high mortality with sudden death when MSCs from PSME4 KO were injected (Figure 6c).

On the basis of the results shown in Figure 6a, we postulated that MSCs from PSME4-KO mice were already affected due to the high level of YAP1. In fact, osteogenic differentiation was greatly accelerated in PSME4-KO MSCs (Appendix A). Hence, delivery of PSME4-KO MSCs was alternatively substituted with rtTA-shPSME4 hTERT-MSCs to clarify the role of PSME4 in the therapeutic efficacy of MSCs. Echocardiographic measurement 2 weeks after injection revealed that control hTERT-MSCs significantly improved both the ejection fraction and the fractional shortening, but PSME4 knockdown hTERT-MCSs failed to improve cardiac function (Figure 6d,e). Cardiac remodeling was further assessed after the mice were euthanized. The total fibrosis area was significantly reduced when control hTERT-MSCs were delivered (Figure 6f,g, first vs. second), whereas this beneficial effect was not observed in the shPSME4-injected group (Figure 6f,h, first vs. third). We directly measured wall thickness of the infarcted zone, which represents chamber remodeling, and we observed that free wall thinning was significantly ameliorated in the control hTERT-MSCs group. However, rtTA-shPSME4-infected hTERT-MSCs were not able to attenuate LV remodeling (Figure 6h). Taken together, our data suggested that acute ablation of YAP1 in the nucleus by the acetylation-dependent proteasome subunit PSME4 was mandatory for the preservation of therapeutic role of MSCs.

## 4. Discussion

Our findings are summarized below (Figure 7). In MSCs, YAP1 is restricted to the cytoplasm in the quiescent status. Apicidin, a potent HDACi, inactivates cytosolic HDAC6, which in turn allows acetylation of YAP1 in the cytosol and stimulates transcription of p21. Acetylation of YAP1 leads to dissociation from the 14-3-3 molecule that governs cytosolic restriction. Acetyl-YAP1 is shuttled into the nucleus and transiently activates its target genes. Robustly activated p21 arrests the transcription of YAP1 and PSME4 degrades acetyl-YAP1 simultaneously. The transient presence of YAP1 in the nucleus triggers the cardiac marker gene expression.

YAP1 was originally highlighted as a functional activator of the Hippo signal, which determines organ growth and size in mammals [27]. Constitutive activation of YAP1 during development enlarges various tissues; however, prolonged activation of YAP1 in the adult period exacerbates tumorigenesis [28]. YAP1 is, thus, regarded as a proto-oncogene, and overexpression of YAP1 is closely linked with poor prognosis of several solid tumors [29,30,31]. YAP1 also governs cardiac development and the repair process after organ damage. Gain of function of YAP1 in the fetal heart stimulates cardiomyocyte proliferation and maturation in vivo, and YAP1-dependent hyperplasia is still effective in postnatal heart [32]. In addition to the proliferation of cardiomyocytes in the developmental stage, YAP1 modulates cardiac disease progression in adult heart. Pressure overload induced by transverse aortic constriction increases total YAP1 and nuclear YAP1. Cardiac-specific deletion of YAP1 results in failure to adapt to pressure overload, which in turn accelerates heart failure transition with massive fibrosis [33]. YAP1 also allows cardiomyocyte survival after MI. Cardiac-specific YAP1 expression facilitates proliferation of cardiomyocyte without any changes in apoptosis of cardiomyocytes [34]. As shown by Lin et al., cardiac function and survival were greater in a murine MI model in which YAP1 was delivered [34].

Although overexpression of YAP1 improves cardiac dysfunction after MI, YAP1 can exacerbate fibrotic changes. Selective overexpression of YAP1 in cardiac fibroblasts promotes proliferation of fibroblasts and differentiation into a mature form called myofibroblasts. Furthermore, specific deletion of YAP1 in the fibroblast results in less fibrosis after MI [35]. Overall, YAP1 orchestrates opposing roles in fibrotic changes after ischemic heart disease: increased survival and contractile function in cardiomyocytes, and clonal expansion and the acceleration of fibrosis in cardiac fibroblasts. Targeted approaches to overcome possible detrimental outcomes will be necessary.

MSCs are multipotent cells that can differentiate into adipocytes, chondrocytes, and osteocytes [36]. ESCs and iPSCs are pluripotent cells that differentiate into cardiomyocytes with high efficiency [37]. On the other hand, MSCs rarely differentiate into cardiomyocytes even under cardiogenic conditions. In our previous report, we found that MSCs were committed to the cardiac lineage within 24 h by apicidin treatment via YAP1–KLF4–miR-130a signaling [17]. In MSCs, YAP1 is high in the hierarchy that governs specification of the cardiac lineage.

One of the most well-established modifications of YAP1 is phosphorylation. Phosphorylation of YAP1 results in cytoplasmic restriction. LAT1/2 initially phosphorylates S127 of YAP1, which triggers sequential phosphorylation at S61, S109, S164, and S397 [22,38]. Phosphorylation including S127 or S397 primes recognition of phosphor-YAP1 by the 14-3-3 molecule, which restricts it in the cytoplasm. Phosphorylation also determines YAP1 fate. Casein kinase 1 preferentially phosphorylates S381 of YAP1, and consequent phosphorylation is developed at S400 and S403, which in turn forms a “phosphodegron”. SCFβ-TRCP recognizes the phosphodegron and facilitates polyubiquitination-dependent proteasomal degradation [38].

Lysine methylation also regulates YAP1 activity. SET1A monomethylates nuclear-imported YAP1 at K342, which interferes with recognition of CRM1, a nuclear export enzyme. K342-methylated YAP1 activates its target gene expression, and prolonged activation of YAP1 in the nucleus accelerates tumorigenesis [39]. In addition to methylation-dependent nuclear tethering, cytoplasmic retention is regulated by methylation. Cytosolic methyltransferase Set7 monomethylases YAP1 at K494, which restricts YAP1 in the cytoplasm [40]. A dense population strongly forces cytoplasmic retention of YAP1, whereas a sparse condition facilitates nuclear localization. However, the methylation-dead mutant of YAP1 is localized in the nucleus even in the confluent environment.

We delineated the role of acetylation in the nuclear targeting and activity of YAP1. Target amino acids for acetylation were confirmed by use of YAP1 containing point mutations. It is interesting that dual modification was possible at a single lysine: methylation and acetylation. We utilized point mutant protein (2KA or 2KR) for the study and the variants were also methylation-resistant [40]. Indeed, we tested localization of YAP1 2KR-EGFP both in the sparse and overconfluent environments and found that notable populations of cells were EGFP positive (Figure 5c). However, no cells were positive when confluent, which indicated that either methylation or acetylation is not a decisive modification for nuclear translocalization but rather a triggering step for another bona fide modification to determine subcellular distribution. Interestingly, the 2KR-EGFP signal was randomly observed, but the density in the nucleus was brighter than that of WT-EGFP, whereas cytoplasmic density was similar (data not shown). We thought that the brighter signal of 2KR-EGFP in the nucleus was the result of delayed clearance due to PSME4-targeted acetyl-YAP1. Once YAP1 is localized in the nucleus, the half-life of nuclear YAP1 is dependent on its acetylation status.

As expected, we observed accumulation of YAP1 in the nucleus when PSME4 was decreased (Figure 5 and Figure 6). We also introduced primarily cultured BM-MSCs from PSME4-null mice to clarify the role of nuclear YAP1 in the MSCs, but the survival rate of mice was strikingly deteriorated when compared PBS-injected control group; the mice died in a few days, which implicated that the PSME-lost MSCs actively worsen disease prognosis (Figure 6). PSME4 is a functional component of the nuclear proteasome that recognizes acetylated substrates [21]. However, the role of PSME4 not clearly known; it might remove harmful components in the MCSs to maintain its stemness. In fact, we also observed that BM-MSCs from PSME4 KO mice preferentially differentiated into the osteogenic lineage (Appendix A). We could not obtain convincing data whether the delivered BM-MSCs were differentiated into osteogenic cell in the mice heart due to early death of the mice; thus, atypical deterioration after PSME4 ablation in the MSCs in vivo should be investigated further.

We are interested in cardiac commitment of MSCs to differentiate mature cardiomyocytes in the heart, as it is still controversial whether the preconditioned MSCs successfully ameliorate contractile impairment of failing heart as functional cardiomyocytes [16]. In fact, it is repeatedly reported that the differentiation rate of delivered MSCs into cardiomyocytes is quite low even though the functional improvement is significant [41]. Hence, many research groups investigated the novel role of MSCs in tissue injury [42,43]. It is noteworthy that MSCs are able to produce numerous molecules in response to an inflammation environment to minimize cell damage [44]. The MSCs in injured tissue also generate and secrete as “secretomes” [45]. Secretomes include soluble proteins, such as cytokines, chemokines, growth factors, and proteases, and extracellular vesicles, such as exosomes and microvesicles [45]. Most secretomes function via paracrine effects, but it remains undefined whether secretomes could modulate host cells in an endocrine manner [46]. The functional importance of secretomes became clearer when the conditioned media were challenged. Many groups that tested only conditioned-media were able to recapitulate the MSC delivery [47,48,49,50]. They solely collected media from MSC cultures and injected them intravenously to MI mice [49,50]. Surprisingly, cell-free media can reduce myocardial death, infarction area, and wall thinning [49,50]. Functional involvement of MSCs in angiogenesis and cardiomyocyte salvage are important parameters of stem cell-based cardiac regeneration but most of the benefits might come from the paracrine effect of secretomes. The significance of the MSC secretome, such as angiopoietin-like 4, is a promising candidate for novel concepts of systolic heart failure from myocardial infarction [51].

## 5. Conclusions

In conclusion, our study showed that, in MSCs, YAP1 is restricted in the cytoplasm in the quiescent status. Apicidin, a potent HDACi, inactivates cytosolic HDAC6, which in turn allows acetylation of YAP1 in the cytosol and stimulates transcription of p21. Acetylation of YAP1 leads to dissociation from the 14-3-3 molecule that governs cytosolic restriction. Acetyl-YAP1 is shuttled into the nucleus and transiently activates its target genes. Robustly activated p21 arrests the transcription of YAP1, and PSME4 degrades acetyl-YAP1 simultaneously.

## Figures and Tables

**Figure 1 pharmaceutics-14-01659-f001:**
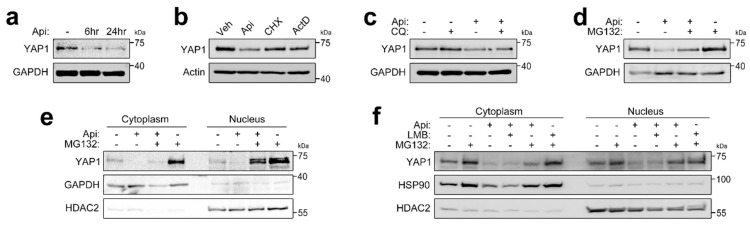
Apicidin prompts nuclear localization and subsequent nuclear degradation of YAP1. (**a**) Apicidin initiates acute loss of YAP1 as early as 6 h after treatment. (**b**) Overnight treatment with 3 μM apicidin (Api) in hTERT-MSCs (human mesenchymal stem cells immortalized by human telomerase reverse transcriptase) robustly reduced the YAP1 protein level while a notable amount of YAP1 remained after treatment with cycloheximide (CHX, 20 μg/mL) or actinomycin D (ActD, 10 μg/mL), which indicates that apicidin actively accelerates YAP1 clearance. (**c**) Apicidin reduced endogenous YAP1 even in the presence of chloroquine (CQ, 10 μM). The clearance of YAP1 was not linked to the lysosomal pathway. (**d**) YAP1 underwent proteasomal degradation predominantly upon apicidin treatment. The proteasome inhibitor MG132 (25 μM, 4 h) significantly attenuated acute loss of YAP1 by apicidin treatment. (**e**) Acute degradation of YAP1 by apicidin stimuli occurred in the nucleus. Overnight incubation with apicidin reduced YAP1 protein in both cytoplasm (first vs. second lane) and nucleus (fifth vs. sixth lane). MG132 predominantly preserved YAP1 degradation in the nucleus (seventh lane), which suggested that the nuclear proteasome pathway might participate in YAP1 clearance. GAPDH represents the cytoplasmic fraction, and HDAC2 shows the nuclear fraction. The cytoplasmic fraction was obtained with hypotonic buffer (0.5% IGEPAL, 10 mM HEPES pH 7.9, 10 mM KCl, and 1 mM DTT), while the nuclear fraction was prepared with hypertonic buffer (1% IGEPAL, 20 mM HEPES pH 7.9, 0.4 M NaCl, 1 mM DTT). (**f**) Apicidin triggers nuclear redistribution of YAP1. Simultaneous treatment with leptomycin B (LMB, 37 nM), a nuclear export inhibitor, and apicidin failed to further accumulate YAP1 in the nucleus (ninth vs. 10th lane, eighth vs. 12th lane). Furthermore, LMB did not reduce YAP1 in the cytoplasm (third vs. fourth lane, second vs. sixth lane). YAP1 provoked unidirectional relocalization of YAP1 from cytoplasm to nucleus.

**Figure 2 pharmaceutics-14-01659-f002:**
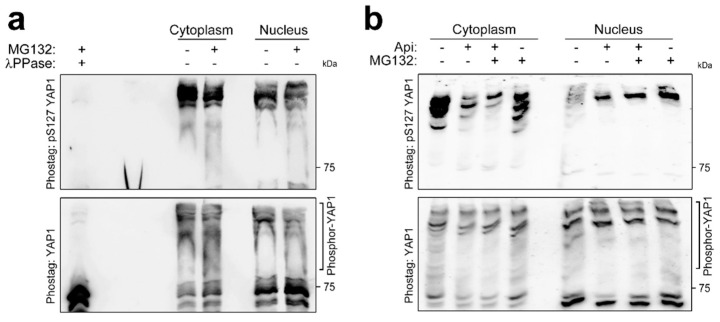
Serine 127 phosphorylation fails to restrict YAP1 in the cytoplasm. (**a**) Phosphor-S127 YAP1 localized both in the cytoplasm and in the nucleus in hTERT-MSCs. Phos-tag phosphoprotein gel stain greatly exaggerated retardation of phosphorylated proteins in the SDS-PAGE. Proteins were separated by various grades of phosphorylation (bottom). Phosphor-specific antisera against S127 phosphor-YAP1 successfully probed nuclear YAP1 accumulated by MG132 treatment. The specificity of phosphor-S127 antibody was confirmed by the first lane, which was incubated with lambda phosphatase; phosphor-S127 antibodies did not detect anything (top). (**b**) Apicidin forced nuclear localization of YAP1 in the presence of S127 phosphorylation. Phosphor-S127 YAP1 antibody visualized nuclear YAP1. Abbreviations: Api, apicidin; λPPase, lambda phosphatase.

**Figure 3 pharmaceutics-14-01659-f003:**
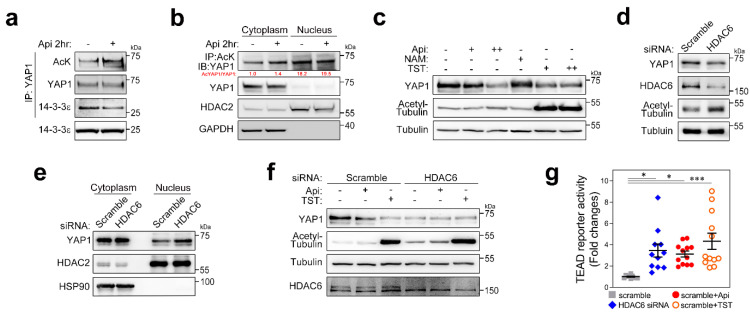
Acetylation governs YAP1 shuttling into the nucleus. (**a**) Apicidin induced acetylation on YAP1. Short-term (2 h) incubation of YAP1 augmented the acetylation level of YAP1 without alteration of YAP1 amounts, which allowed dissociation of YAP1 from the 14-3-3 molecule that tethered YAP1 in the cytoplasm. (**b**) YAP1 acetylation in response to 2 h of treatment with apicidin took place in the cytoplasm, but most acetyl-YAP1 was enriched in the nucleus. HDAC2 shows the nuclear fraction, whereas GAPDH represents cytoplasmic components. (**c**) Cytosolic HDAC(s) might participate in YAP1 acetylation. Tubastatin A phenocopied apicidin-dependent YAP1 degradation. Overnight treatment with Tubastatin A successfully reduced endogenous YAP1 as strongly as did treatment with apicidin, while nicotinamide failed to do so. Acetyl-tubulin delineated cytosol HDAC inhibition. Note that apicidin also induced tubulin acetylation slightly. (**a**,**e**) HDAC6 regulated YAP1 acetylation. Three days of knocking down with siRNA against HDAC6 significantly reduced YAP1 in the hTERT-MSCs (**d**), while 1 day of incubation after siRNA transfection accelerated nuclear localization of YAP1 (fourth lane) (**e**). HDAC2 depicts the nuclear fraction and HSP90 the cytoplasmic fraction. (**f**) HDAC6 mainly governs the fate of YAP1. Acute degradation of YAP1 was further tested in the absence of HDAC6. Both apicidin and Tubastatin A degraded YAP1, which was not observed when HDAC6 was removed (fourth, fifth, and sixth lane). HDAC6 siRNA itself reduced YAP1, but further loss of YAP1 by either apicidin or Tubastatin A was not detected. (**g**) Acetylation dynamics of YAP1 potentiated TEAD reporter activity. Nuclear translocalization by acetylation of YAP1, HDAC6 siRNA, apicidin, or Tubastatin A activated the TEAD reporter, which represents the Hippo signal. One-way analysis of variance with Tukey’s honestly significant difference (HSD) post hoc test. * *p* < 0.05, *** *p* < 0.001. Abbreviations: AcK, acetyl-lysine; NAM, nicotinamide; TST, Tubastatin A.

**Figure 4 pharmaceutics-14-01659-f004:**
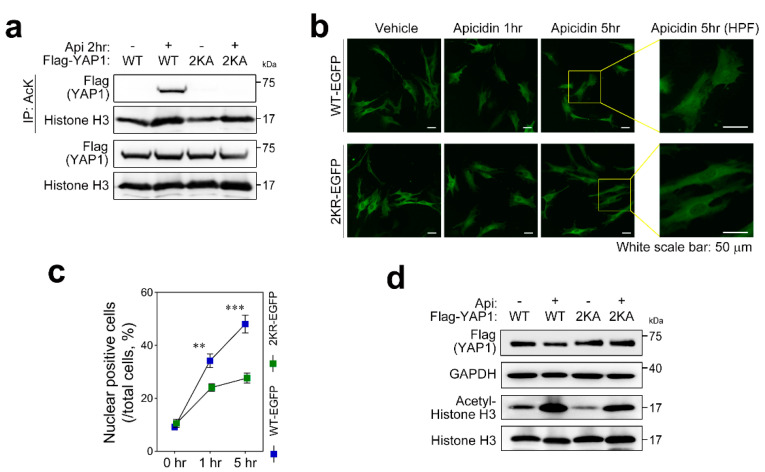
Lysine 494/497 are acetylation targets of YAP1. (**a**) Acetylation of YAP1 was not detected when two lysines, 494 and 497, were mutated (2KA). Output with histone H3 after immunoprecipitation with acetyl-lysine was used for internal control. Acetyl-histone H3 delineated intracellular effect of apicidin treatment. (**b**,**c**) Acetylation facilitated nuclear localization. Enhanced green fluorescent protein (EGFP) was fused at the C terminus of YAP1 to visualize location of YAP1. Wildtype YAP1-EGFP started to accumulate in the nucleus as early as 1 h after treatment with apicidin, which was greatly increased after 5 h. Nuclear targeting of the acetyl-dead form of YAP1 was significantly lower than that of wild-type. HPF, high-power field. (**d**) Acetyl-dead mutant of YAP1 remained intact even in the presence of apicidin. Simultaneous treatment with apicidin and cycloheximide failed to degrade acetylation-resistant mutant of YAP1, while wildtype YAP1 was cleared. The gray circle indicates vehicle (or tet-off) condition, and the red rhombus denotes doxycycline (or tet-on) treatment. (**c**) Two-way ANOVA and Tukey’s HSD multiple comparison were performed. ** *p* < 0.01, *** *p* < 0.001.

**Figure 5 pharmaceutics-14-01659-f005:**
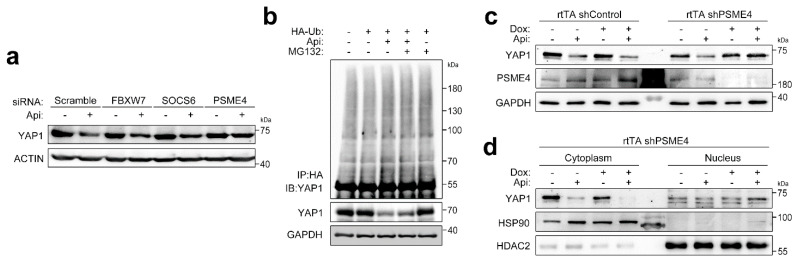
PSME4 degrades YAP1 in a polyubiquitination-independent manner. (**a**) Proteasome activator subunit 4 (PSME4) participates in YAP1 degradation in apicidin-treated cells. siRNA against F-box protein (FBXW) 7 or suppressor of cytokine signaling (SOCS) 6 failed to preserve acute loss of YAP1. A notable amount of YAP1 was visualized when PSME4 was targeted. (**b**) Apicidin did not induce YAP1 polyubiquitination. MG132 (25 μM, 4 h) successfully mitigated YAP1 degradation but failed to induce YAP1 polyubiquitination. (**c**) PSME4 shRNA attenuated YAP1 degradation (seventh vs. eighth lane). (**d**) Preserved YAP1 mediated by PSME4 knockdown mainly accumulated in the nucleus (sixth vs. eighth lane). Gray circle: vehicle. Red square: apicidin. Two-way ANOVA with Tukey’s HSD post hoc test. Abbreviations: Ctrl, control. DOX, doxycycline. Ub, ubiquitin.

**Figure 6 pharmaceutics-14-01659-f006:**
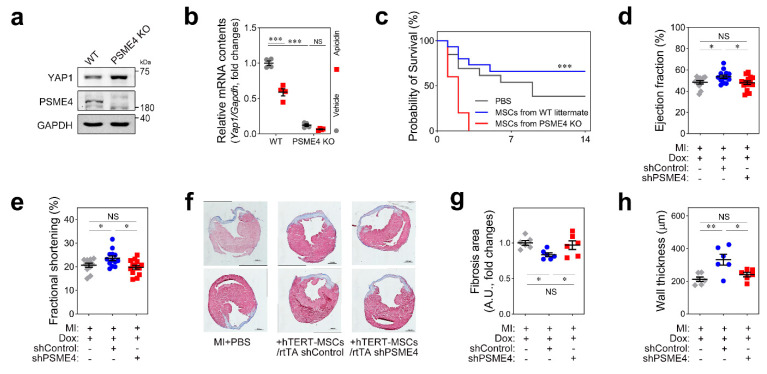
PSME4 governs MSC differentiation via YAP1. (**a**) YAP1 protein was more abundant in primary cultures of MSCs from PSME4-null individuals compared with wildtype littermates. (**b**) Aberrant transcription of YAP1. YAP1 mRNA was significantly lowered in PSME4 knockout (KO) bone marrow (BM) MSCs. (**c**) Survival rate after direct injection of MSCs either from PSME4 KO or wildtype littermate was measured for 2 weeks. When MSCs from PSME4 KO mice were injected, survival was paradoxically decreased. Statistical significance was obtained by applying Mantel–Cox log-rank test. Cohort size: PBS *n* = 13, MSCs from WT littermate *n* = 15, MSCs from PSME4 KO *n* = 10. (**d**,**e**) Cardiac function, ejection fraction (**d**), and fractional shortening (**e**) were not improved in rtTA-shPSME4 MSCs delivered group. Cohort size: PBS *n* = 11, MSCs/rtTA shControl *n* = 13, MSCs/rtTA shPSME4 *n* = 14. (**f**–**h**) Cardiac remodeling was not impaired. Cardiac fibrosis (**g**,**h**) and wall thickness after ischemic insults (**h**) were not improved in the group with PSME4-knockdown hTERT-MSCs injection. Gray rhombus: PBS injection group. Blue circle: rtTA-shControl hTERT-MSCs injection group. Red square: rtTA-shPSME4 hTERT-MSCs delivered group. One-way ANOVA with Tukey’s HSD multiple comparison test. * *p* < 0.05, ** *p* < 0.01, *** *p* < 0.001. NS, not significant.

**Figure 7 pharmaceutics-14-01659-f007:**
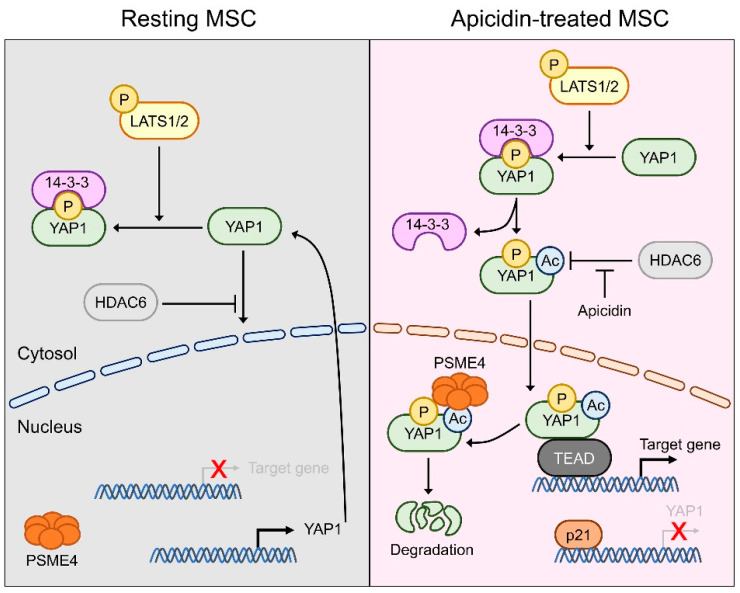
Working hypothesis. Hippo pathway in the resting MSC: LATS1/2 phosphorylated YAP1 S127. 14-3-3 recognizes phosphor-YAP1 and restricts YAP1 in the cytoplasm. HDAC6 indirectly interferes with nuclear translocalization of YAP1 (**left**). Acute regulation of YAP1 occurs under apicidin treatment. Apicidin blocks cytosolic HDAC6 and facilitates acetylation indirectly. Apicidin transactivates p21, which induces transcriptional arrest of YAP1. Acetylation of YAP1 dissociates YAP1 from 14-3-3 regardless of phosphorylation status. Acetylated-YAP1 is shuttled into the nucleus after being released from the cytosolic tethering molecule, 14-3-3. Nuclear YAP1 initiates target gene expression and recognition by PSME4, a subunit of the nuclear proteasome. PSME4 degrades acetyl-YAP1 in the nucleus (**right**).

## Data Availability

Not applicable.

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
