# Peer review of "PSME4 Degrades Acetylated YAP1 in the Nucleus of Mesenchymal Stem Cells"

_pharmaceutics, 2022, doi:10.3390/pharmaceutics14081659_

Round 1
Reviewer 1 Report
PSME4 degrades acetylated YAP1 in the nucleus of mesenchymal stem cells to induce cardiac commitment
The authors wrote an interesting article for the role of PSME4 degrading acetylated YAP1 in MSCs to induce cardiac commitment, however I have some comments regarding the submitted manuscript. The comments are as follows:
Comments
1. Could the authors show characterization of their MSC in accordance to the minimum criteria for MSCs as set by The International Society for Cellular Therapy (ISCT) position statement (Dominici et al 2006). The MSC has not been tested for: MSC must express CD105, CD73 and CD90, and lack expression of CD45, CD34, CD14 or CD11b, CD79alpha or CD19 and HLA-DR surface molecules.
2. Similarly, this extends to showing their MSC can differentiating to osteoblasts, adipocytes and chondroblasts in vitro as per The International Society for Cellular Therapy (ISCT) position statement (Dominici et al 2006). If the data is present – I would suggest to place it into their supplemental section.
3. Has the authors considered showing how this is translated in using primary BM-MSCs instead of immortalized hTERT-MSCs? hTERT-MSCs can often result in mutations in the long term. It would support these findings further if some experiments were performed with primary BM-MSCs?
I understand the concerns with using shRNA in hTERT-MSCs as this is long-term suppressing expression. It would support the manuscript if the authors would incorporate siRNA with primary MSCs.
4. Could the authors comment on whether the source of MSCs matter in their experimental disease model?
5. Figure 3g. The * and *** is not clear enough. Suggest to bolden or make the stars bigger for reader to see.
6. The authors should mention in the discussion that the MSC-secretome can also show a functional capacity in driving a therapeutic effect.
The papers (I would suggest to reference these in the manuscript) below shows examples of the MSC-secretome (primarily exosomes or extracellular vesicles) that can restore diseased organs in various models and not exclusively MSCs (the cells themselves):
· Nikfarjam, S., Rezaie, J., Zolbanin, N.M. et al. Mesenchymal stem cell derived-exosomes: a modern approach in translational medicine. J Transl Med 18, 449 (2020). https://doi.org/10.1186/s12967-020-02622-3
· Taglauer, Elizabeth S et al. “Antenatal Mesenchymal Stromal Cell Extracellular Vesicle Therapy Prevents Preeclamptic Lung Injury in Mice.” American journal of respiratory cell and molecular biology vol. 66,1 (2021): 86-95. doi:10.1165/rcmb.2021-0307OC.
· Taglauer, Elizabeth S et al. “Mesenchymal stromal cell-derived extracellular vesicle therapy prevents preeclamptic physiology through intrauterine immunomodulation†.” Biology of reproduction vol. 104,2 (2021): 457-467. doi:10.1093/biolre/ioaa198
· Reis, Monica et al. “Mesenchymal Stromal Cell-Derived Extracellular Vesicles Restore Thymic Architecture and T Cell Function Disrupted by Neonatal Hyperoxia.” Frontiers in immunology vol. 12 640595. 15 Apr. 2021, doi:10.3389/fimmu.2021.640595
· McKay, Tina B et al. “Extracellular Vesicles in the Cornea: Insights from Other Tissues.” Analytical cellular pathology (Amsterdam) vol. 2021 9983900. 22 Jul. 2021, doi:10.1155/2021/9983900
· Willis, Gareth R et al. “Mesenchymal stromal cell-derived small extracellular vesicles restore lung architecture and improve exercise capacity in a model of neonatal hyperoxia-induced lung injury.” Journal of extracellular vesicles vol. 9,1 1790874. 13 Jul. 2020, doi:10.1080/20013078.2020.1790874
· Willis, Gareth R et al. “Extracellular Vesicles Protect the Neonatal Lung from Hyperoxic Injury through the Epigenetic and Transcriptomic Reprogramming of Myeloid Cells.” American journal of respiratory and critical care medicine vol. 204,12 (2021): 1418-1432. doi:10.1164/rccm.202102-0329OC
Overall, the article provides a well-established experimental outline and an interesting insight on the role of PSME4 degrading acetylated YAP1 in MSCs to induce cardiac commitment. I have some comments regarding the MSC-secretome in driving the therapeutic response as well. But other than that – this would enhance the pharmaceutics manuscript.
Reviewer 2 Report
The study of Yoon et al. evaluates the impact of the treatment with Apicidin in YAP1. Authors observed that Apicidin treatment induced acetylation of YAP1, which resulted in nucleus translocation and degradation by acetylation-dependent proteasome subunit, thus affecting YAP1 availability and activity. In addition, authors claimed that acute ablation of YAP1 in the nucleus is mandatory for cardiac commitment of MSCs.
Overall, the manuscript is focused on the impact of MSCs in CVDs and MSCs cardiac regenerative potential and the impact of the YAP1 ablation to stimulate this effect. However, the results presented do not support this line of thought and the conclusions formulated by the authors. It seems that authors fitted the general effect observed, which was thoroughly studied, to the specific cellular model used (MSC) and the idea of cardiac commitment, which was unsubstantiated.
Major Issues:
1. Authors reference the positive effects of MSCs for the treatment of CVDs. There are also many other studies showing there is no effect or even a negative effect. In fact, authors mention [line 66-71] one of those studies.
2. Nonetheless, assuming only the positive studies, MSC properties for tissue regeneration and in particular for differentiation into cardiomyocyte-like cells still present a highly contested idea. The capacity of MSCs to originate a population of cardiomyocytes or cardiac cells by direct differentiation is, at best, in dispute. At best, one can argue, there is a small potential for transdifferentiation. Unfortunately, most articles studying this commitment potential rely on the use of unspecific factors such as 5-Aza (epigenetic modifier) and provide little to no evidence of MSC commitment into actual functional cells. In fact, the idea of using MSCs as a source of cells to repopulate the injury myocardium has been largely shifted towards their well-known and accepted immunomodulatory properties (secretome) as the likely mechanism of action for the cardio protective effect observed in some of the studies.
3. Nonetheless, one can focus on the results showed in the current manuscript. Authors based all the claims of cardiac commitment in a few fold-changes of the cardiac markers NKX2.5, TNNI3 and GATA4 using different conditions and experiments. There is no other evidence of cardiac commitment or specificity. The fold change observed is very small compared with the ones usually seen in cardiac progenitors or early cardiomyocytes versus undifferentiated cells. Nevertheless, assuming it would be enough, authors did not assess if this apparently increase in gene expression resulted in an increase and protein and, more important, if those proteins would actually result in lineage commitment of some type, which is the relevant question and the only one that could start to fundament the claims postulated by the authors. Specifically, what is happening at the cellular level? Are MSCs actually committing towards cardiac cells? Are there observable changes in the cell population? Does the MSCs actually originate subpopulations of cardiac cells? Authors did not evaluate any effect past the changes in relative gene expression and assumed that this was a direct reflection of cardiac commitment, which is insufficient for the claim that there was any type of MSC cardiac commitment observed by the degradation of acetylated YAP1 by PSME4.
Reviewer 3 Report
In apicidin-treated MSCs, YAP1 undergoes acetylation by inhibiting cytoplasmic HDAC6, dissociation of 14-3-3 from YAP1, shuttling into the nucleus to activate target genes and proteasome-dependent degradation in the nucleus by PSM4, inducing cardiac commitment.
Very interesting manuscript but I recommend summarizing these described mechanisms in graphical form, which would allow a better understanding for the reader.
-line 22 - clarify this sentence
-Lines 49-50 - include references
-Provide the reference for the protocol of immortalization of the MSCs/characterization of the cells.
- provide all the original western blot membranes for the replicates in a file.
- fig 4b (apidicin 5h WT-EGFP) is too blurry to see YAP1 in the nucleus.
- Line 563 - "Cardiac-specific YAP1 expression facilitates the proliferation of cardiomyocyte without any changes in apoptosis of cardiomyocytes" provide reference
Reviewer 4 Report
Yoon and colleagues describe that PSME4 degrades acetylated YAP1 in the nucleus of mesenchymal stem cells to induce cardiac commitment.
Overall, the manuscript is well written, the design of the experiments is logic, and the quality of the data is convincing. There are only some small points to consider:
Line 66: “Phase 3 CHART-1 trial” –A reference should be included. The paragraph is overall misleading as significant benefit was only observed in the subgroup with extreme cardiac dilation.
Throughout the manuscript: The term “cardiac commitment” is overstating. Please describe which parameters were significantly different expressed.
The observed lethality in PSME4 knockout mice with MI should be discussed. One possibility is that PSME4 has other functions in the heart besides degrading YAP1. Endogenous mesenchymal stem cells might naturally contribute to repair and depend on PSME4 and finally, PSME4 will degrade plenty of substrates besides YAP1.
Round 2
Reviewer 1 Report
The authors have amended all changes and I believe it is now suitable for publication.
Author Response
We greatly appreciate reviewer's encouragement.
Reviewer 2 Report
I understand the authors point of view and I would like to commend the effort of the authors in the rebuttal to the comments made. Nonetheless, authors mentioned that “In this study, we tried to focus on the regulation mechanism of YAP1 protein rather than stem cell-based therapy.”
But as I commented before one of the major issues was that the manuscript was written exactly contrary to that: “Overall, the manuscript is focused on the impact of MSCs in CVDs and MSCs cardiac regenerative potential and the impact of the YAP1 ablation to stimulate this effect.” Plus “It seems that authors fitted the general effect observed, which was thoroughly studied, to the specific cellular model used (MSC) and the idea of cardiac commitment, which was unsubstantiated.” Thus, the argument of this manuscript being focused only on the regulation mechanisms of YAP1 seems to be contrary to what I read in the current manuscript. Therefore, my issues with it still remain unchanged.
Regarding the issue that authors named “3. Data Quality”. In my point of view the authors were still unable to address the key questions “what is happening at the cellular level? Are MSCs actually committing towards cardiac cells? Are there observable changes in the cell population? Does the MSCs actually originate subpopulations of cardiac cells?”. And I would humbly suggest the authors try to address those in their other works that are currently in preparation since I would absolutely be one of the first readers.
Wish authors the best for future endeavors,
Author Response
We deeply appreciate reviewer’s suggestion. We understand and accept common concerns raised by both reviewer and editor. We removed all the data and related descriptions regarding “cardiac commitment” from main text and supplementary dataset. Current version is limitedly focused on acute regulation mechanism in response to apicidin treatment; acetylation-dynamics and the role of PSME4 in terms of nuclear clearance of YAP1. Discussion is also revised accordingly. Every single change is highlighted in the yellow. We are grateful to the reviewer for the constructive suggestions, and we hope to continue our further study to figure out the key questions you mentioned.
We hope the current revision will be now suitable for publication.
Round 3
Reviewer 2 Report
I found the compromise from the authors enough and have no further comments.